# Mining folded proteomes in the era of accurate structure prediction

**Charles Bayly-Jones**[1,2]*, **James C. Whisstock**[1,2]*

**1** Department of Biochemistry and Molecular Biology, Monash University, Clayton, Australia, **2** Biomedicine Discovery Institute, Faculty of Medicine, Nursing and Health Sciences, Monash University, Clayton, Australia

* charles.bayly-jones@monash.edu (CB-J); james.whisstock@monash.edu (JCW)

## Abstract

Protein structure fundamentally underpins the function and processes of numerous biological systems. Fold recognition algorithms offer a sensitive and robust tool to detect structural, and thereby functional, similarities between distantly related homologs. In the era of accurate structure prediction owing to advances in machine learning techniques and a wealth of experimentally determined structures, previously curated sequence databases have become a rich source of biological information. Here, we use bioinformatic fold recognition algorithms to scan the entire AlphaFold structure database to identify novel protein family members, infer function and group predicted protein structures. As an example of the utility of this approach, we identify novel, previously unknown members of various pore-forming protein families, including MACPFs, GSDMs and aerolysin-like proteins.

## Author summary

Virtually every cellular process in all organisms on Earth is driven by molecular nano-machines known as proteins. The diverse functions of proteins are the result of the unique three-dimensional shape adopted by a given protein molecule. It is therefore important to determine the shape of a given protein, which unlike DNA and our genes, cannot be known from its sequence alone. Since two proteins with similar shapes typically have a similar function, knowing a protein shape provides crucial clues about its function. By virtue of decades of experimental work and advances in artificial intelligence, this complex shape can now be computationally predicted for any protein whose composition is known. Scientists have used these and other methods to produce enormous libraries of protein shapes consisting of nearly a million unique entries. However, these libraries are too large and too complex for researchers to 'read'. We use shape-comparison algorithms to carefully check these shape-libraries to gain insight into the potential function and biological role of previously unknown proteins. Furthermore, we identified new members of protein families using this technique. We show that shape-matching algorithms and computationally generated shape-libraries can be used effectively together to yield new insights and expedite scientific endeavours.

**Data Availability Statement:** All supplementary data is made available as a public submission to zenodo: https://zenodo.org/search?page=1&size=20&q=5893808.

**Funding:** JCW acknowledges funding from the Australian Research Council, the Australian Research Data Commons, and the National Health and Medical Research Council of Australia. The funders had no role in study design, data collection and analysis, decision to publish, or preparation of the manuscript.

**Competing interests:** The authors have declared that no competing interests exist.

## Introduction

Knowledge of a proteins' structure is a powerful means for the prediction of biological function and molecular mechanism [1,2]. Accordingly, powerful pairwise fold recognition tools such as DALI [3] have been developed that permit searching of known fold space in order to identify homology between distantly related structurally characterised proteins. These approaches can identify homologous proteins even when primary amino acid sequence similarity is not readily detectable. This method is particularly useful when a protein of no known function can be flagged as belonging to a well characterised fold class (e.g. Rosado et al. [4]).

A key and obvious limitation of using fold recognition to infer function is that the structure of the protein of interest needs to first be determined. By virtue of advances in experimental techniques and judicious deposition of results over several decades, this limitation is now being addressed by machine learning approaches. Today, in the era of accurate protein structure prediction [5,6], it is possible to build a reasonably accurate library comprising representative structures of all proteins in a proteome [7–9] (Fig 1A, 1B and 1C). One utility of such a resource, is that fold recognition approaches for prediction of function can now be applied to any protein (Fig 1D and 1E).

To investigate the utility of this approach we used established bioinformatic tools to mine the "foldome" (S1 Text). We employ the popular DALI algorithm due to its sensitivity and robustness. We constructed a locally hosted DALI database of all protein structures predicted

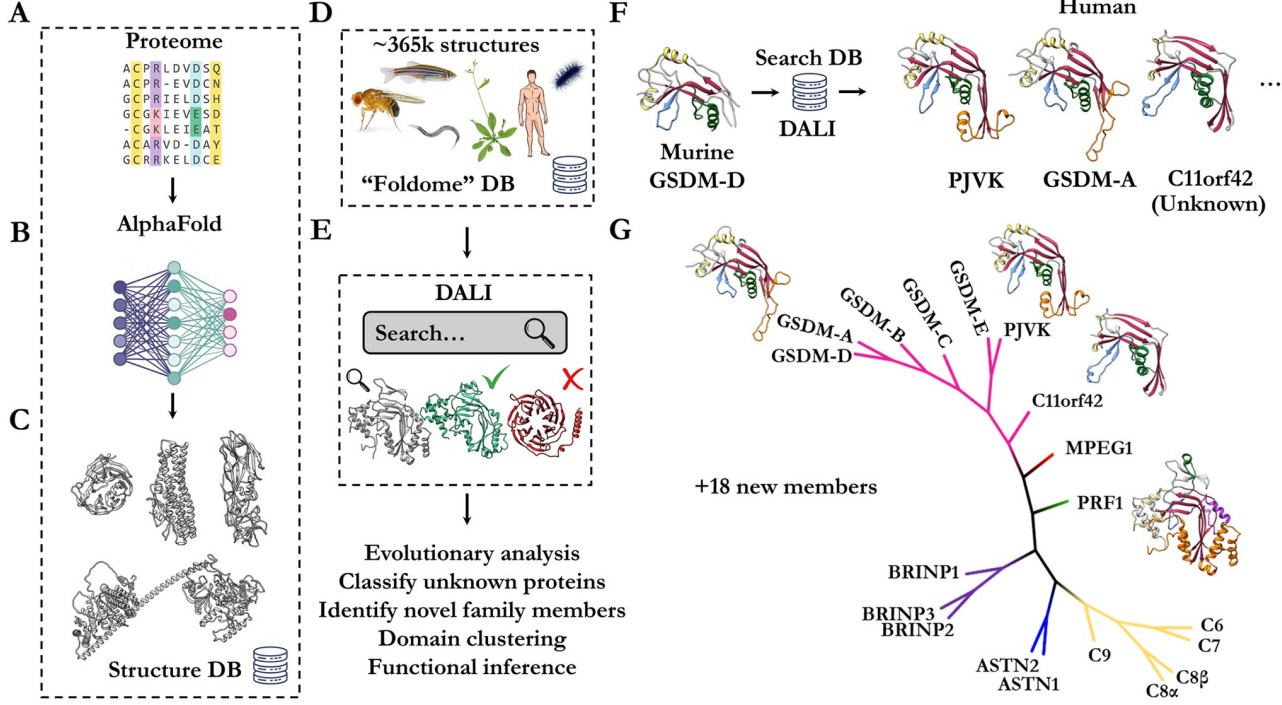

**Fig 1. Conceptual overview of structure-guided fold recognition against the AlphaFold database. a.** Entire proteome sequence databases are converted by (**b**) machine-learning methods into high-accuracy (**c**) structural model predictions. **d.** Curation of these structural databases into a unified resource of structures (S1 Text). **e.** DALI based fold matching to perform functional inference, identification of unknown members and structural classification. **f.** Searching the foldome in (**d**) for matches of murine GSDM-D N-terminal domain yields MACPF/GSMD family members, including C11orf42 –an unknown member of the GSDM family. **g.** Example phylogenetic analysis of perforin-like proteins identified from the foldome by DALI. Alignment of the central MACPF/GSDM fold is possible by extracting domain boundaries based on AlphaFold prediction allowing specific comparison between members without interfering ancillary domains.

by AlphaFold, covering humans to flies to yeast (Fig 1D and 1E). We then began mining the whole database using a probe structure representing a well characterised protein superfamily (in this case the perforin-like superfamily of pore forming immune effectors). Repeating this search with different MACPF probes (e.g., MPEG1, perforin, C9) yields very similar results, indicating DALI is robust to the chosen search template.

## Design and implementation

### Generation and acquisition of AlphaFold models

All AlphaFold models were obtained from the EMBL EBI database (https://alphafold.ebi.ac.uk/) for each available model organism. For any searches where existing models were not available from the PDB, these were generated using AlphaFold hosted through ColabFold [10]. A regex search of PDB metadata for 'uncharacterised' was used to curate a subset of uncharacterised or unknown proteins in the human foldome. Atomic coordinates for these files were subsequently discarded if their pLDDT score was less than 70. Remaining models which possessed fewer than 100 residues were discarded.

### Construction of local DALI search engine and database

All DALI searches were performed using DaliLite [3] (v5; available from http://ekhidna2.biocenter.helsinki.fi/dali/) on one of two Linux workstations equipped with 16-core or 20-core Intel i7 CPU and 128 Gb of DDR4 RAM. DALI database was generated as described in the DALI manual. Briefly, for every AlphaFold model a randomised four character internal "PDB" code was generated and associated with the model (*unique_identifiers.txt*). Subsequently, all models were imported and converted to DALI format to enable structure all-versus-all searches. Individual proteomes were isolated as separate lists of entries or combined, to enable independent or grouped searches.

### Construction of local SA Tableau (GPU accelerated) search engine and database

All SA Tabelau searches were performed on the same Linux workstations as for the DALI searches; however jobs were split equally into four sets and each set was executed on a single Nvidia GTX1080 GPU. Since SA Tableau was originally written and compiled [11] for outdated CUDA architectures, we re-compiled SA Tableau under modern CUDA (v8.0), gcc (v4.9.3) and g++ (v5.4.0). To run SA Tableau, it was necessary to create a conda environment with python2.7, where numpy (v1.8.1) and biopython (v1.49) were found to properly execute and run the original code. SA Tableau databases and distance matrices were calculated with '*buildtableauxdb.py*' and combined into ASCII format with '*convdb2.py*'(available from http://munk.cis.unimelb.edu.au/~stivalaa/satabsearch/ at the date of publication). SA Tableau results were sorted and selected based on expectation value with a cut-off of $1 \times 10^{-4}$.

### Proteome-wide assignment of Pfam

In order to search the entire Pfam classification against a structural proteome database, we used the GPU-accelerated SA Tableau search algorithm to expedite the search process. Furthermore, we selected to search only the *S. aureus* foldome as this represents the smallest and therefore most computationally inexpensive example proteome. All Pfam classifications were represented by structure in one of two ways [12]. Firstly, trRosetta models for ~7,000 Pfam classifications were recently produced and used without modification. Secondly, of the remaining 60% of entries, we used the ProtCID database [13] (http://dunbrack2.fccc.edu/

ProtCiD/default.aspx) to link Pfam IDs to known protein structures in the PDB. These exemplar structures from the PDB were downloaded and single chains were extracted from each model (that is, only a single copy of each domain was considered). Domain boundaries and chain IDs defined by ProtCID were used to discard unrelated chains and residues that did not pertain to the particular Pfam classification in question. Finally, the trRosetta and exemplar structures were searched against the entire *S. aureus* foldome with SA Tableau.

## Results

This approach readily yields functional insight into previously uncharacterised proteins (Fig 1F, 1G **and** S1 Table). For example, structure-based mining identified all known perforin / GSDM family members, but also identified a likely new member of the GSDM pore-forming family in humans, namely C11orf42 (uniprot Q8N5U0—a protein of no known function). Remarkably there is only 1% sequence identity between the GSDMs and C11orf42 despite predicted conservation of tertiary structure. In humans, C11orf42 is expressed in testis and is highly expressed in thyroid tumours [14]. Moreover, CRISPR screens [15–17] identified C11orf42 as contributing to fitness and proliferation in lymphoma, glioblastoma and leukaemia cell lines (BioGRID gene ID 160298) [18].

Identification of C11orf42 as a likely GSDM family member permits several useful predictions. Owing to the presence of a GSDM fold, we postulate that C11orf42 may share GSDM-like functions such as oligomerisation and membrane interaction. Unlike other GSDMs [19,20], however, inspection of the predicted structure suggests that C11orf42 lacks membrane penetrating regions entirely. These data imply that C11orf42 may have lost the ability to perforate lipid bilayers and instead may function as a scaffold of sorts, as has been postulated for members of the perforin superfamily [21].

We next expanded our analysis to all proteomes covering 356,000 predicted structures; these computations take ~24 hours on a 16 CPU Intel i7 workstation. We identified roughly 16 novel perforin-like proteins across the twenty-one model organisms covered by the Alpha-Fold database (Fig 1G, S1 Table **and File 1 in** https://zenodo.org/record/5893808#.YiE_LOhKhPY). Domain boundaries defined by the structure prediction were identified manually and the perforin/GSDM-like domains were aligned based on fold. We constructed a phylogenetic tree based on the structure-constrained multiple sequence alignment (MSA), suggesting that C11orf42 is potentially related to the precursor of the common GSDMs. Curation of sequences based on predicted structures, such as this, may enable further, more comprehensive evolutionary analyses. For example, by using newly identified structural homologs as seed sequences for iterative PSI-BLAST searches of sequence databases or by studying the gene loci of newly discovered family members.

We next decided to perform these foldome-wide searches for several other pore-forming protein families, identifying new members of aerolysins, lysenins, cry1 toxins and more (S1 Fig, S1 Table **and File 1 in** https://zenodo.org/record/5893808#.YiE_LOhKhPY). Members of these toxin families have applications in next-generation sequencing (both DNA/RNA [22,23] and polypeptide [24–27]), as well as agricultural applications in crop protection. We anticipate the new members of these families to be of utility in translational research programs. Remarkably, many of the hits we identified suggest the unexpected presence of pore-forming protein families that were previously thought to be entirely absent in the selected phyla–for example aerolysin-like proteins in *Drosophila*, *C. elegans*, yeast and zebrafish.

To further assess the utility of the database in functional inference, we curated a subset of the human proteome corresponding to uncharacterised proteins of unknown function (Fig 2A, 2B and 2C). These proteins are largely unannotated, lacking both domain and

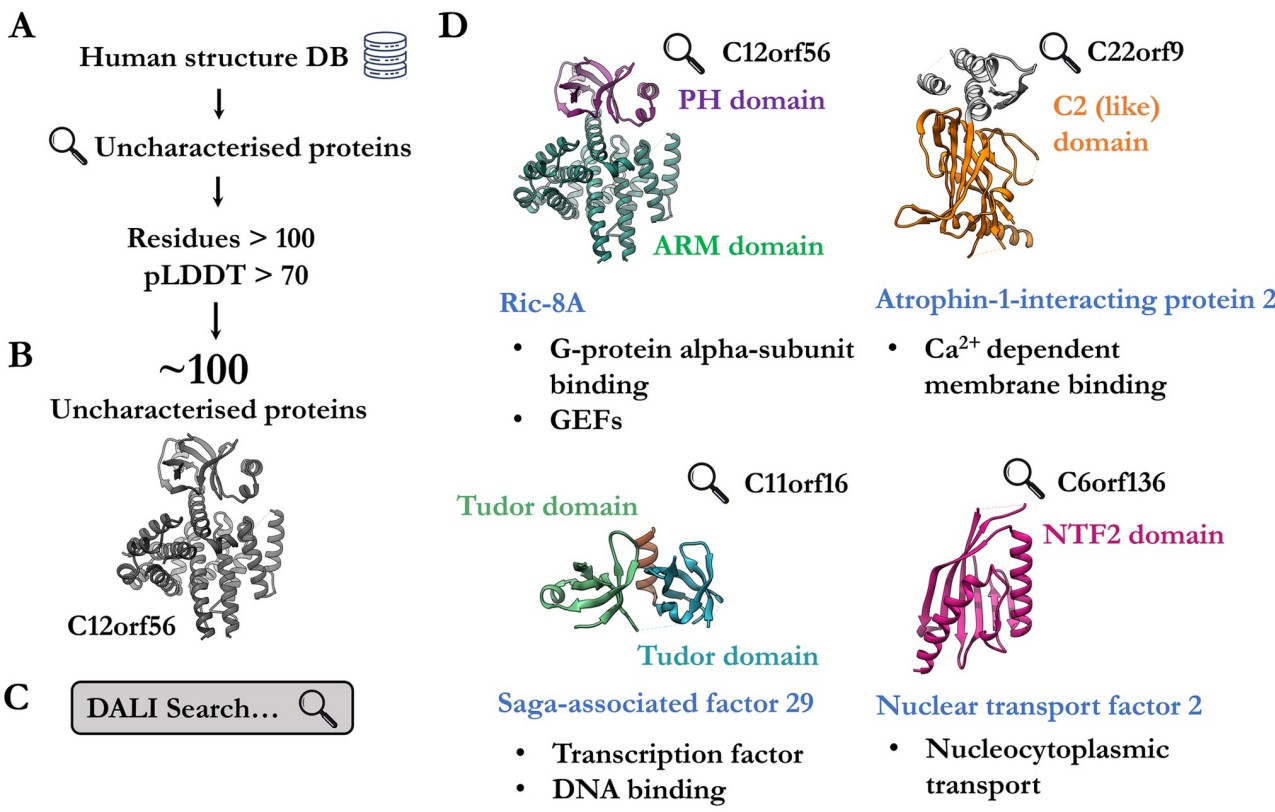

**Fig 2. Identification of uncharacterised human proteins by fold-recognition. a.** Uncharacterised human proteins were curated from the AlphaFold database. **b.** Low-confidence regions of AlphaFold models were excluded based on pLDDT criteria. **c.** All models were screened against the rest of the human AlphaFold database. **d.** Four examples (C12orf56, Q8IXR9; C22orf9, Q6ICG6; C11orf16, Q9NQ32; C6orf136, Q5SQH8) of uncharacterised proteins where fold-matching enabled the assignment of domain composition (labelled in various colours). Furthermore, homologs or similar proteins (blue label) provide insight into potential function (black dot points).

functional descriptions. We pruned all regions of the predicted structures to have pLDDT (per-residue confidence score) greater than 70 and discarded models for which fewer than 100 residues remained. These became the probe structures for iterative searches against the whole human foldome to identify known proteins with assigned domains and function. We provide these as **File 2 in** https://zenodo.org/record/5893808#.YiE_LOhKhPY for the convenience of the reader.

From these analyses, we highlight four notable examples of uncharacterised structures which met the criteria and yielded insight into potential function (Fig 2D). One of these, C12orf56, appears to be a previously unknown GTPase activator protein. When compared to its homolog Ric-8A [28], the PH domain appears to sterically occlude binding of Gα proteins and may result in a potentially autoinhibited conformation. Previously, the identity of this protein was most likely obscured in sequence-based approaches due to the abnormally large loop insertion in the PH domain (Fig 2D). Similarly, a putative nuclear import factor (NFT) with strong homology to NFT2 was identified. These examples demonstrate the utility of structure-guided curation and annotation of uncharacterised proteins. Unlike domain assignment by primary sequence analysis, fold-matching algorithms are sensitive and robust [29–31]. We anticipate that domain assignment by fold-matching will likely provide more accurate and informative predictions over existing sequence analysis methods, especially in contexts where

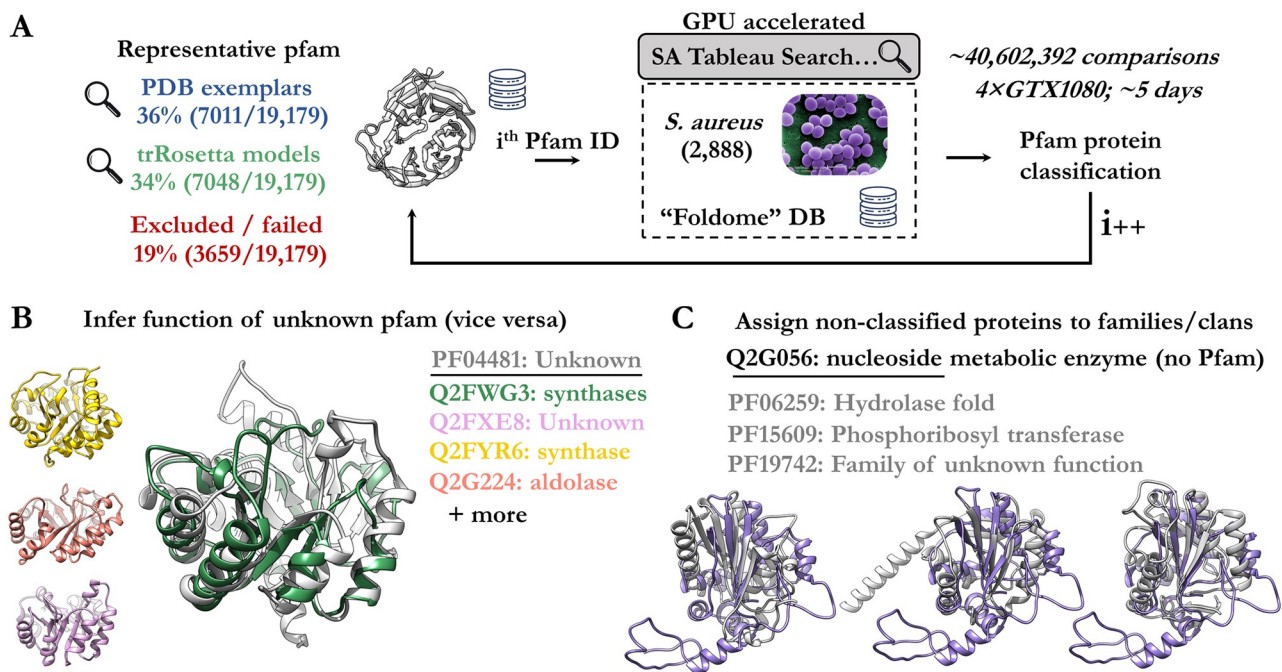

**Fig 3. Proteome-wide search and classification of Pfam groups. a.** Overview of analysis, firstly search models representing pfam entries were curated from either trRosetta or the PDB. These were each searched against the entire *S. aureus* foldome to identify matches. The final output was filtered by expectation value. **b**. Select examples of *S. aureus* enzymes classified into Pfam category PF04481. This Pfam category previously had unknown role or function. **c**. The uncategorised protein (Q2G056) is homologous to different Pfam groups with similar enzymatic function (PF06259, PF15609, PF19742).

sequences have poor overall homology or possess discontinuous breaks and insertions. Of course, imputed function remains to be experimentally validated.

Lastly, many conserved protein families have no known function and roughly a quarter of sequences in the proteome are not assigned a protein family [12]. Further, domain annotations for numerous proteins are incomplete. In an attempt to employ structure matching to assign domain composition or identify protein families, we searched representative protein domains against the *Staphylococcus aureus* foldome (for simplicity we chose the smallest available foldome) to score unknown and known proteins according to their similarity (Fig 3A). These representative structures comprise the entire trRosetta Pfam library and curated exemplar structures from the PDB for Pfam entries not modelled by trRosetta. The public nature of these representative structures makes the trRosetta models a convenient choice, however, computing AlphaFold predictions for each Pfam entry would likely give an improved representative library. A remaining 19% of Pfam entries were excluded (fewer than 50 residues, absence of PDB entry).

Here, we employed a GPU-accelerated fold recognition software, SA Tableau search [11], to expedite the large comparison which was not computationally tractable with DALI. The analysis outputs a ranked list of all proteins which match the query domain (**File 3 in** https://zenodo.org/record/5893808#.YiE_LOhKhPY). As such these serve as first-pass approximation of structure-assigned domain annotation and family classification for the *S. aureus* proteome. We provide a full mapping of pfam to *S. aureus* entries ranked by likelihood, which enables prediction of function by similarity (S2 Table). Examples of the results include the prediction that the unknown Pfam group (PF04481) is structurally related to a group of synthases

(Fig 3B) and the classification of the *S. aureus* protein (Q2G056) as a hydrolase-like or transferase-like fold.

Exhaustive searches of predicted structures serve as a sensitive, but computationally expensive, domain and family assignment tool for proteins which lack sequence annotations or where domain assignment has not been successful using sequence-based approaches. Owing to prohibitively time-consuming computational limitations, it was not feasible for us to search the entire foldome across all organisms. The dedication of high-performance computing resources to the remaining proteomes, or particularly subsets that are still unknown, may be merited. Similarly, the AlphaFold database would benefit from application of other structure-based classification methods (such as adaptations to classification schemes of SCOPe [32] or ECOD [33]). The curated subset of PDB entries used for DALI searches are available as a resource to expedite efforts by others (**File 3 in** https://zenodo.org/record/5893808#.YiE_LOhKhPY) and supplement the trRosetta Pfam models (publicly available: http://ftp.ebi.ac.uk/pub/databases/).

Finally, it is our perspective that adoption of the above analyses among structural biologists may be beneficial. Common practice of firstly searching for related folds before beginning a project, may accelerate investigations and improve the likelihood of success. For example, one might first generate an accurate structural prediction of the target molecule, then search this against larger foldome databases (via DALI [3] or FoldSeek [34] webserver) to gain insight into function and putative mechanism. In this way, before experimentation, previously obtained knowledge of function can provide rationale, guide inquiry and minimise unnecessary or resource intensive efforts–saving time and money. Likewise, the identification of homologs in model organisms may facilitate parallel studies in vivo or in situ.

## Future directions

The need for modern and sensitive implementations of fold-matching algorithms has once again become relevant. The current release of AlphaFold predictions has expanded the available structural database by more than double, with additional contributions expected to reach nearly a million entries in the near future. Exhaustive search algorithms, such as DALI, are slow and scale poorly, meaning searching structural databases of these sizes is not tractable. As such improvements and further work on fold matching algorithms are paramount to enable rapid searching and exploration of these new resources (for example, FoldSeek [34]). Other algorithms for fold matching are notably much faster than DALI, such as FoldSeek (comparable sensitive to DALI) [34] and 3D Zernike moment decomposition of protein structures and subsequent k-means nearest neighbour [35]. However, the latter comes with sensitivity trade-offs. A mixed approach may be beneficial, where 3D Zernike descriptors could be initially used to filter the database, followed by an exhaustive DALI search. This would be akin to a coarse initial pass to define a smaller subset of the search space to allow a computationally tractable exhaustive local search. Alternatively, sequences could be first filtered based on exclusion criteria, such as length.

Overall, the efficacy of structure mining depends on the accuracy of predicted models. Currently, this is dependent on MSAs for the detection of evolutionary covariance, however new single-sequence structure prediction methods are emerging that do not rely on sequence alignment [36]. Currently, the extent and quality of the MSA will affect the quality of AlphaFold/RoseTTAFold predictions and thus the quality of search results. Notably, protein families with extensive primary sequence conservation may not benefit from structure-guided mining, as existing techniques are likely sufficiently sensitive, as well as being far quicker and more computationally efficient. As such, protein folds that are structurally conserved but have poor overall sequence conservation may represent ideal targets for structure-based mining.

## Supporting information

**S1 Fig. Select examples of newly identified PFPs. a.** Identified MACPFs from slime mold and zebrafish. **b.** Various β-PFPs with aerolysin-like pore-forming domains which resemble aerolysin, lysenin, epsilon toxin, monalysin and LSL. Observed in numerous organisms including drosophila, *C. elegans*, zebrafish, yeast, among others. **c.** Several novel α-haemolysin-like proteins identified in *S. aureus* and plants. The putative receptor binding domain is coloured green and the pore-forming domain is coloured grey with the expected transmembrane region in red.
(TIF)

**S1 Table. Curated lists of pore-forming proteins identified by DALI search of AlphaFold database, organised by query.**
(XLSX)

**S2 Table. Ranked list of structurally-assigned pfam matches against the *S. aureus* foldome.**
(XLSX)

**S1 Text. Definition of a "foldome".**
(DOCX)

## Acknowledgments

CBJ acknowledges helpful discussion and feedback from Dr. Bradley A. Spicer and Dr. Andrew Ellisdon.

## Author Contributions

**Conceptualization:** Charles Bayly-Jones, James C. Whisstock.

**Formal analysis:** Charles Bayly-Jones, James C. Whisstock.

**Investigation:** Charles Bayly-Jones.

**Methodology:** Charles Bayly-Jones.

**Resources:** James C. Whisstock.

**Supervision:** James C. Whisstock.

**Visualization:** Charles Bayly-Jones.

**Writing – original draft:** Charles Bayly-Jones.

**Writing – review & editing:** Charles Bayly-Jones, James C. Whisstock.

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
