## [Decision Letter · Decision Letter 0]

24 Nov 2021

Dear Dr. Bayly-Jones,

Thank you very much for submitting your manuscript "Mining folded proteomes in the era of accurate structure prediction" for consideration at PLOS Computational Biology. As with all papers reviewed by the journal, your manuscript was reviewed by members of the editorial board and by several independent reviewers. The reviewers appreciated the attention to an important topic. Based on the reviews, we are likely to accept this manuscript for publication, providing that you modify the manuscript according to the review recommendations.

Importantly, please provide a command line tool or ideally a web server to perform the structural similarity search and analysis from the paper.

Sincerely,

Dina Schneidman

Software Editor

PLOS Computational Biology

[LINK]

Specifically, please provide a command line tool or ideally a web server to perform the structural similarity search and analysis from the paper.

Reviewer's Responses to Questions

**Comments to the Authors:**

Reviewer #1: The manuscript presents a case study on mining the computationally predicted structures in the recently released AlphaFold Protein Structure Database for structural similarities. The authors use established fold recognition algorithms to compare proteins from the entire database to known members of several pore-forming protein families and succeed in identifying previously unknown members of such families.

The authors' methods represent a clever and straightforward-to-apply workflow to explore computationally predicted structures. The paper explains well why the particular results from their examples are of biological interest. Beyond corroborating the approach itself, these results demonstrate that computationally predicted protein structures can be used to discover relevant connections and relations between different proteins that cannot be obtained from sequence databases.

This study is the first of its kind (as far as I am aware) which comes with a lot of merit but also necessitates some additional tests and discussion to form a solid basis for future similar ventures. In particular, the following points should be addressed:

- It is not clear to me whether using a different reference protein from one family would lead to the same result. For example, if instead of GSDM-D one had searched for similarities to a MACPF, would the results have included C11orf42, too? Similarly, could one identify all MACPF/GSDM members by using C11orf42 as a reference?

- When expanding the analysis to all proteomes (line 52ff), the computational cost increases strongly. Could one in principle alleviate this problem with a less expensive pre-screen that excludes proteins that, by some simpler criteria, clearly cannot have a similar fold?

- The authors introduce the term "foldome" for a set of structures corresponding to a proteome. According to their definition (one of many potential folds per sequence), a proteome has multiple potential foldomes. This definition leaves no room to describe a set of structural ensembles, which would intuitively be associated with the term foldome meaning "entire set of folds."

- While the "era of accurate structure prediction" is indubitably owed to the machine learning techniques, as much credit should be attributed to advances in the experimental techniques that determine the structures from which these models learn.

- The paper alludes to "more comprehensive evolutionary analyses" in line 59. It would help to elaborate on that or provide a few examples.

Reviewer #2: Summary of the paper

The authors point out a fairly obvious use-case for ML predictions of protein structure and then describe some interesting applications. Structures predicted by AlphaFold2 or similar neural networks are queried using fold recognition tools such as DALI in order to find structural homologs with very low sequence identity and impute function to uncharacterized proteins. The primary contributions are:

- The authors identify new pore-forming proteins, including a new human perforin / GSDM with only 1% sequence homology to known examples

- The authors query human proteins with confident structural predictions against the remainder of the AF2 human foldome, identifying possible functions for several uncharacterized proteins

- The authors query predicted structures for Pfam domains of unknown function against the S. aureus AF2 foldome

Main Review

Strengths

In general, the paper is concise and clear.

I appreciated the discussion of limitations and possible improvements

The example of structural homology with only 1% sequence homology is impressive!

Major Weaknesses

I think the contribution would be stronger if the authors also made it easier (with a command-line tool or web server) for scientists with less-specialized knowledge to do structural homology search against AF2 predictions.

Minor weaknesses

The paper could be stronger with justifications or reasoning for tool choices. Why use DALI vs SA Tableau for different tasks? Or why use trRosetta predictions instead of AF2 or RosettaFold predictions for some tasks?

Likewise, how were the foldomes chosen? Why limit searches to the human foldome or the S. aureaus foldome? Would it be better to search against a deduplicated, "representative" predicted structure library? If not, why not?

While acknowledging that bench verification of imputed functions is out of scope for this paper, I would like some discussion of the trustworthiness of functions imputed via structural homology, even under the assumption that the structure predictions are accurate.

Given that AF2 predicts accurate protein structures given MSAs, could we do homology search by just comparing MSAs?

Summary of the review

While the idea of applying structural homology search to ML structure predictions is fairly obvious, the authors do it carefully and find convincing, biologically-interesting results. Together, this makes a strong case for making this sort of search a part of the standard toolbox when working with proteins of unknown function, and the work is of broad interest to protein biologists and engineers. Overall, this paper is suitable publication in PLOS Comp Bio with minor revisions.

**Have the authors made all data and (if applicable) computational code underlying the findings in their manuscript fully available?**

Reviewer #1: Yes

Reviewer #2: Yes

PLOS authors have the option to publish the peer review history of their article (what does this mean?). If published, this will include your full peer review and any attached files.

Reviewer #1: No

Reviewer #2: **Yes: **Kevin Kaichuang Yang

Figure Files:

Data Requirements:

Reproducibility:

References:

---

## [Decision Letter · Decision Letter 1]

16 Feb 2022

Dear Dr. Bayly-Jones,

We are pleased to inform you that your manuscript 'Mining folded proteomes in the era of accurate structure prediction' has been provisionally accepted for publication in PLOS Computational Biology.

Best regards,

Dina Schneidman

Software Editor

PLOS Computational Biology

Please address the comment of Reviewer 2 in the final version

Reviewer's Responses to Questions

**Comments to the Authors:**

Reviewer #1: My questions and concerns have been sufficiently addressed.

Congratulations on the impressive results!

Reviewer #2: The authors respond thoroughly to the concerns from the first set of reviews. My one remaining concern is about this paragraph:

"Unlike domain assignment by primary sequence analysis, fold-matching algorithms are sensitive and

robust. We anticipate that domain assignment by fold-matching will likely provide more accurate and

informative predictions over existing sequence analysis methods, especially in contexts where

sequences have poor overall homology or possess discontinuous breaks and insertions. Of course,

imputed function remains to be experimentally validated."

Intuitively, it makes sense that fold-matching would be more sensitive and robust than primary sequence analysis. However, citing examples from the literature where these are compared head-to-head would strengthen this section.

**Have the authors made all data and (if applicable) computational code underlying the findings in their manuscript fully available?**

Reviewer #1: Yes

Reviewer #2: Yes

PLOS authors have the option to publish the peer review history of their article (what does this mean?). If published, this will include your full peer review and any attached files.

Reviewer #1: **Yes: **Martin Vögele

Reviewer #2: **Yes: **Kevin Kaichuang Yang

---

## [Editor Report · Acceptance letter]

22 Mar 2022

PCOMPBIOL-D-21-01823R1 

Mining folded proteomes in the era of accurate structure prediction

Dear Dr Bayly-Jones,

I am pleased to inform you that your manuscript has been formally accepted for publication in PLOS Computational Biology. Your manuscript is now with our production department and you will be notified of the publication date in due course.

With kind regards,

Livia Horvath
